# Glecaprevir–pibrentasvir for chronic hepatitis C: Comparing treatment effect in patients with and without end-stage renal disease in a real-world setting

**Hsu-Heng Yen**[1,2,3]*, **Pei-Yuan Su**[1], **Ya-Huei Zeng**[1], **I-Ling Liu**[1], **Siou-Ping Huang**[1], **Yu-Chun Hsu**[1], **Yang-Yuan Chen**[1], **Chia-Wei Yang**[1], **Shun-Sheng Wu**[1], **Kun-Ching Chou**[1]*

**1** Division of Gastroenterology, Department of Internal Medicine, Changhua Christian Hospital, Changhua, Taiwan, **2** Institute of Medicine, Chung Shan Medical University, Taichung, Taiwan, **3** General Education Center, Chienkuo Technology University, Changhua, Taiwan

☯ These authors contributed equally to this work.
* 91646@cch.org.tw, blaneyen@gmail.com (HHY); 84798@cch.org.tw (KCC)

**Data Availability Statement:** All relevant data are within the manuscript and its Supporting Information files.

## Abstract

### Introduction

Chronic hepatitis C virus (HCV) infection is increasingly observed in patients with renal disease. With the introduction of glecaprevir/pibrentasvir (GLE/PIB) as a pan-genotype therapy for HCV, treatment efficacy is expected to rise.

### Materials and methods

This retrospective study evaluated the efficacy and safety of GLE/PIB treatment in adults with HCV infection and end-stage renal disease (ESRD). The primary end point was sustained virological response (SVR) observed 12 weeks after completed treatment.

### Results

We enrolled 235 patients, including 44 patients with ESRD. Median age was 60 years, and 48% were males. Twenty-two percent had cirrhosis. HCV genotypes 1 (43%) and 2 (41%) were the most common. The overall SVR rate was 96.6%. Patients with ESRD were older than those without (67.6 years *vs* 58.3 years, p < 0.001) and trended toward having a higher prevalence of cirrhosis (32% *vs* 19%, p = 0.071). A significant proportion of patients with ESRD complained of skin itching during treatment (61% *vs* 26%, p < 0.001), and the SVR rate were similar between these two groups (95.45% *vs* 96.86%, p = 0.644).

### Conclusions

Despite a higher rate of pruritus among patients with ESRD, GLE/PIB-based therapy achieved similarly high SVR rates among patients with and without ESRD.

**Funding:** HHY received research funds from Changhua Christian Hospital (108-CCH-IRP-018 and 109-CCH-IRP-008).

**Competing interests:** The authors have declared that no competing interests exist.

## Introduction

Uncontrolled chronic hepatitis C virus (HCV) infection eventually leads to liver cirrhosis or liver cancer and results in significant morbidity and mortality [1]. It has been estimated that 130–170 million people are infected with HCV, resulting in a global prevalence of 2%–3% [2, 3]. Over the past two decades, interferon-based therapy has been the standard treatment for HCV infection. Sustained virological response (SVR) rates in patients with HCV infection range from 60%–80% following six months of peg-interferon-based therapy [4, 5]. Recently, HCV treatment was revolutionised with the introduction of direct-acting antiviral (DAA) therapy. High SVR rates (> 90%) and good treatment tolerance have been reported for interferon-free regimens. These new agents have been reimbursed by the Taiwanese health care system since 2017, and the government has set the goal of obtaining an 80%-treatment coverage rate with DAAs by 2025 [6–8]. In the past, treatment success for HCV infection in patients with chronic kidney disease lagged behind that of the general population [9]. Today, it is possible to cure most HCV-infected patients with oral DAAs.

Glecaprevir (GLE) is a NS3 protease inhibitor, and pibrentasvir (PIB) is a NS5A inhibitor. Since August 2018, glecaprevir–pibrentasvir (GLE/PIB) has been reimbursed in Taiwan for patients [1] with confirmed HCV viremia with no prior DAA therapy. Both GLE and PIB are excreted through the biliary tree and fecal route. There is no need for dosage adjustment for patients with renal function impairment. It is the first and only pan-genotype DAA for both patients with and without end-stage renal disease (ESRD) [2]. In the EXPEDITION-4 study [2], a high SVR rate and good safety profile were reported for patients with ESRD. Until now, real-world data on the safety and effectiveness of GLE/PIB in patients with hepatitis c infection with and without ESRD are still limited. Thus, we report our real-world experience with anti-HCV therapy with GLE/PIB in our institution.

## Materials and methods

### Materials

This retrospective study included DAA treatment naïve patients undergoing treatment for HCV infection, who received anti-HCV therapy with GLE/PIB between August 2018 and December 2019 at Changhua Christian Hospital. The study was approved by the Changhua Christian Hospital Institutional Review Board (CCH IRB No 190814), and documentation of informed consent was waived because the study was conducted retrospectively. Medical information, including demographics, baseline medical conditions, anti-HCV treatment regimen and duration, laboratory values and information on adverse events were extracted from electronic patient records. The anti-HCV treatment response at the end of treatment was compared with that after the treatment. All procedures were carried out in accordance with relevant guidelines and regulations of the Changhua Christian hospital.

### Treatment, efficacy, and safety evaluation

Our primary goal was to compare the treatment response in patients with and without ESRD in our institution. We used ART HCV assays (RealTime HCV and HCV Genotype II, Abbott Molecular, Abbott Park, IL, USA) to quantify HCV RNA concentrations and genotyping. End-of-treatment viral response (ETVR) was defined as an HCV RNA level < the lower limit of quantification (LLOQ) when completing the treatment course. A SVR was defined as an HCV RNA level < LLOQ at 12 weeks after the last medication. GLI/PIB was reimbursed by the National Health Insurance of Taiwan since Aug 2018 for patients with confirmed HCV viremia without prior use of DAA. The treatment period was 8 weeks for patients without liver

cirrhosis and 12 weeks for patients with compensated liver cirrhosis. Virological treatment failure was defined as either (a) non-response: HCV was detected during and at end of treatment; or (b) relapse: HCV was undetectable at the end of treatment but detectable during the follow-up period. The diagnosis of cirrhosis was established by liver biopsy, observing stage-4 fibrosis or by ultrasound-based findings of cirrhosis or the presence of oesophageal varices during endoscopy. Two endpoints for SVR were evaluated. The intention-to-treat group (ITT) included patients receiving at least one dose of GLE/PIB and the per-protocol group (PP), established by excluding patients due to non-virological failure. The rate of premature treatment discontinuation was also analysed. ESRD was defined as chronic kidney disease stage 5 with dialysis [10] in the present study.

## Statistical analyses

Statistical analyses were performed using the SPSS software version 18.0 and Medcalc software version 19.3. Differences between the two groups were considered statistically significant when probability (P) values were < 0.05.

## Results

### General characteristics of patients with HCV infection

A total of 235 patients with HCV infection received GLE/PIB-based anti-HCV therapy during the study period (Fig 1). Forty-four of these patients had ESRD with haemodialysis (n = 40) or peritoneal dialysis (n = 4). Most patients were female (52%) with a median age of 60 years. The ESRD group had a higher proportion of interferon treatment naïve patients. Liver cirrhosis was present in 51 patients (22%). The most common HCV genotype observed was type 1 (43%), followed by type 2 (41%), type 3 (8%) and others (8%). Five percent of the patients were

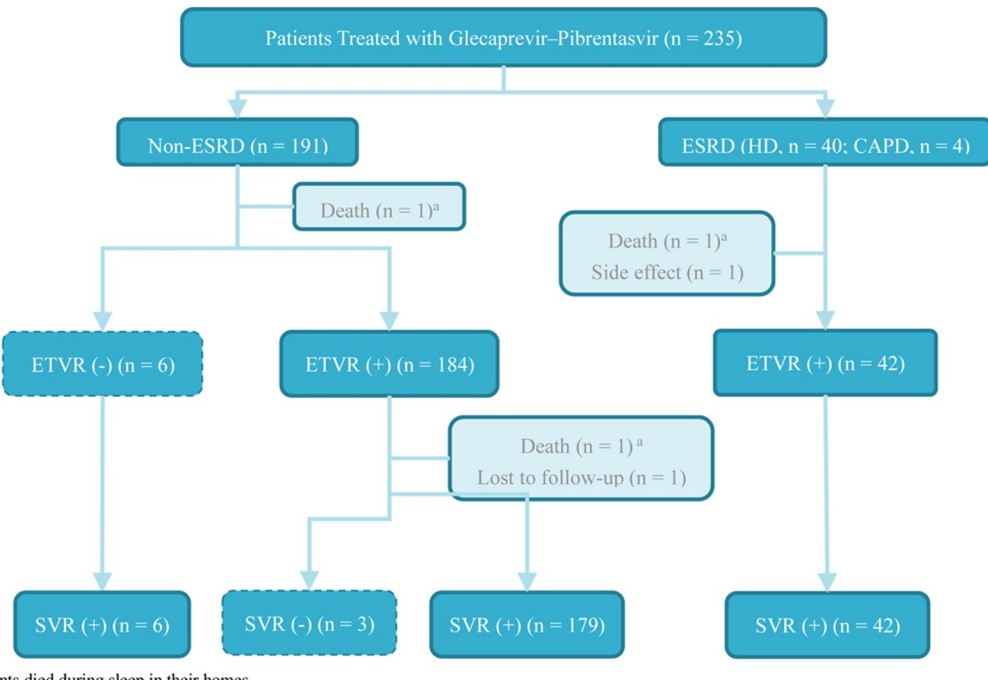

ᵃThese patients died during sleep in their homes.

**Fig 1. Algorithm of patient inclusion.**

co-infected with hepatitis B. Three fatalities were observed during (n = 2) and after (n = 1) the treatment period. All the deaths were judged not to be related to the treatment. Premature treatment termination was observed for one patient due to side effects (skin itching). One patient was lost to follow-up after achieving end-of-treatment response. Six patients in the non-ESRD group without ETVR achieved SVR after completed treatment. Three patients in the non-ESRD group with ETVR exhibited viral relapse 12 weeks after the treatment. One of the patients with drug abuse and HIV co-infection had a different HCV genotype at the time of virological relapse, and so re-infection was suspected. The overall ITT SVR rate was 96.6%, and the PP SVR rate was 98.7%. Patient characteristics are summarised in Table 1 and Fig 1.

## Comparison of patients with/without ESRD

The clinical features and treatment responses of patients with and without ESRD are displayed in Table 2. Patients with ESRD were older than those without ESRD (67.6 years *vs* 58.3 years, P < 0.001). Lower platelet counts, levels of haemoglobin, albumin, GOT/GPT, and HCV viremia were observed in the ESRD group at baseline. Patients with ESRD had a higher Fibrosis-4 (FIB-4) index than those without ESRD and more patients with ESRD had hypertension. The distribution of HCV genotypes did not differ between the two groups. Both patient groups achieved similarly high rates of EVTR and SVR.

## Side effects of the treatment

Pruritus is the most commonly reported side effect of GLE/PIB treatment, followed by insomnia, malaise, and abdominal discomfort (Table 3). A significant increase in the levels of bilirubin, GOT, and GPT was observed in <1.3% of the patients. We further analysed the time course and severity of pruritus and its impact on the SVR (Table 4). Significantly more patients with ESRD reported pruritus during treatment compared with those without ESRD (61% *vs* 26%, P < 0.001). Twenty-three percent of the patients with ESRD experienced grade-3 pruritus. The presence of pruritus occurred mostly during the first 4 weeks of treatment, subsequently the rate of pruritus decreased over time. One patient in the ESRD group discontinued treatment prematurely due to this side effect. The presence of pruritus did not impair the overall SVR rate of patients with and without ESRD.

## Discussion

In this single-centre cohort study of HCV-infected patients receiving GLE/PIB therapy, we report an overall SVR rate of 96.6%. Our results are comparable with those observed in the clinical trial setting [2, 11–13] and other real-world reports [1, 7, 14], identifying GLE/PIB as a highly efficacious treatment regimen for HCV infection. Despite the several disadvantageous factors like older age and a higher prevalence of cirrhosis and pruritus-related side effects during treatment among patients with ESRD, we observed similar rates of SVR in patients with and without ESRD (95.5% *vs* 96.9%, P = 0.644).

One of the issues of treating HCV-infected patient with renal function impairment in the interferon-based therapy era was the side effects [9, 15, 16], and such patients are regarded as a difficult-to-treat population. Dialysis-dependent patients with active HCV infections are at increased risk of liver disease-related mortality, generally have a poor quality of life, and are at increased risk of developing cardiovascular diseases [17]. In addition, dialysis patients with HCV infection constitute a transmission reservoir in dialysis centres, making HCV treatment of high importance for this population. Liu et al [5] reported a 64% SVR rate for ESRD patients treated with interferon and ribavirin combination therapy, with a 7% withdrawal rate mainly due to treatment-related side effects. Despite the fact that untreated HCV-infected dialysis

**Table 1. Patient characteristics at baseline.**

| Characteristic | All patients (N = 235) | Non-ESRD (N = 191) | ESRD (N = 44) | P value |
|---|---|---|---|---|
| Male, n (%) | 112 (48) | 86 (45) | 26 (59) | 0.092 |
| Age (in years), mean ± SD | 60 ± 12.5 | 58.3 ± 12 | 67.6 ± 12.1 | <0.001 |
| FIB-4, median (IQR) | 1.89 (1.29–2.78) | 1.81(1.25–2.6) | 2.33(1.71–3.15) | 0.007 |
| Cirrhosis, n (%) | 51 (22) | 37 (19) | 14 (32) | 0.071 |
| Hepatitis B, n (%) | 11 (5) | 9 (5) | 2 (5) | 1.000 |
| HIV, n (%) | 5 (2) | 5 (3) | 0 | 0.587 |
| Diabetes, n (%) | 32 (14) | 23 (12) | 9 (20) | 0.142 |
| Hypertension, n (%) | 42 (18) | 29 (15) | 13 (30) | 0.025 |
| Stroke, n (%) | 6 (3) | 5 (3) | 1 (2) | 1.000 |
| Cancer, n (%) | 14 (6) | 11 (6) | 3 (7) | 0.730 |
| HCV RNA (log IU/mL), median (IQR) | 5.74 (4.82–6.35) | 5.9 (4.98–6.41) | 5.33 (4.04–6.1) | 0.005 |
| HCVgenotype, n (%) | | | | |
| Type 1[a] | 102 (43) | 78 (41) | 24 (55) | 0.137 |
| Type 2[b] | 97 (41) | 80 (42) | 17 (39) | 0.822 |
| Type 3 | 18 (8) | 17 (9) | 1 (2) | 0.209 |
| Type 4 | 1 (0.4) | 1 (0.5) | 0 | 1.000 |
| Type 6 | 13 (6) | 11 (6) | 2 (5) | 1.000 |
| Mixed genotype[c] | 4 (2) | 4 (2) | 0 | 1.000 |
| Prior Therapy, n (%) | | | | |
| Interferon naïve | 199 (84.7) | 157 (82.2) | 42 (95.5) | 0.028 |
| Interferon relapse | 3 (1.3) | 3 (1.6) | 0 (0) | 1 |
| Interferon failure | 33 (14.1) | 31 (16.2) | 2 (4.5) | 0.045 |
| Complete Treatment, n/N (%) | | | | |
| 8 week course | 181/184 (98) | 152/154 (99) | 29/30 (97) | 0.416 |
| 12 week course | 46/51 (90) | 33/37 (89) | 13/14 (93) | 1.000 |
| Height, cm, mean± SD | 160.1 ± 8.5 | 160.1 ± 8.3 | 160.5 ± 9.3 | 0.760 |
| Weight, kg, median (IQR) | 60(53–69.4) | 59.7(52–70) | 60.3(53–66) | 0.779 |
| BMI, kg/m$^2$, median (IQR) | 23.3(21.3–26) | 23.32(21.3–25.97) | 23.325(20.7–26.21) | 0.924 |
| Creatinine, mg/dL, median (IQR) | 0.87(0.65–1.46) | 0.78(0.63–1.01) | 8.1(7.09–10.23) | <0.001 |
| GOT (AST), U/L, median (IQR) | 37(27–53) | 38(29–58) | 31(21.5–40) | <0.001 |
| GPT (ALT), U/L, median (IQR) | 40(25–61) | 42(27–69) | 25.5(20–39) | <0.001 |
| Platelet,10$^3$/μL, median (IQR) | 193(151–236) | 199(153–239) | 169.5(131–206.5) | <0.001 |
| INR, median (IQR) | 0.94(0.89–0.99) | 0.94(0.89–0.99) | 0.91(0.89–0.97) | 0.209 |
| Bilirubin, mg/dL, median (IQR) | 0.67(0.47–0.88) | 0.7(0.5–0.9) | 0.56(0.435–0.695) | <0.001 |
| Albumin, g/dL, median (IQR) | 4 (3.7–4.3) | 4.1(3.9–4.3) | 3.45(3.1–3.65) | <0.001 |
| Hemoglobin, g/dL, median (IQR) | 13.5(11.4–14.7) | 13.9(12.7–15) | 10.55(9.3–11.7) | <0.001 |

Data are expressed as n (%), median (interquartile range \IQR]), or mean ± standard deviation. Categorical variables were compared using the $\chi^2$ test or Fisher's exact test, as applicable; continuous variables were compared using Student's t test or Mann–Whitney $U$ test. P values <0.05 were considered statistically significant.

[a] HCV genotype 1, including 1, 1a, 1b.

[b] HCV genotype 2, including 2, 2a, 2b

[c] Mixed genotype, including 1b+2, 1b+3, 1 or 6

patients had a 2.62-fold increase in mortality risk compared with treated patients in a recent study from Taiwan [8, 9], only a very low proportion (6.01%) of the patients received antiviral treatment. Despite the decrease in side effects and improved SVR rates since the introduction of interferon-free DAA therapy [12, 14, 18], GLE/PIB is currently the only approved therapy

**Table 2. On-treatment and off-therapy virological responses.**

| HCV RNA < LLOQ[a] | All patients (N = 235) | | Non-ESRD (N = 191) | | ESRD (N = 44) | | P value |
|---|---|---|---|---|---|---|---|
| | n/N (%) | 95% CI | n/N (%) | 95% CI | n/N (%) | 95% CI | |
| **End of Treatment Response (ETVR)** | | | | | | | |
| Intention to Treat | 226/235 (96.2) | 92.9–98.2 | 184/191 (96.3) | 92.6–98.5 | 42/44 (95.5) | 84.5–99.4 | 0.677 |
| Per Protocol Analysis | 226/232 (97.4) | 94.5–99.0 | 184/190 (96.8) | 93.3–98.8 | 42/42 (100) | 91.6–100 | 0.595 |
| **Sustained Response (SVR)** | | | | | | | |
| Intention to Treat | 227/235 (96.6) | 93.4–98.5 | 185/191 (96.9) | 93.3–98.8 | 42/44 (95.5) | 84.5–99.4 | 0.646 |
| Per Protocol Analysis | 227/230 (98.7) | 96.2–99.7 | 185/188 (98.4) | 95.4–99.7 | 42/42 (100) | 91.6–100 | 1.000 |
| **Reason for non-SVR, n** | | | | | | | |
| Death | 3 | | 2 | | 1 | | 0.465 |
| Discontinued due to side effect | 1 | | 0 | | 1 | | 0.187 |
| Lost to follow-up | 1 | | 1 | | 0 | | 1.000 |

Statistical analysis for one proportion was performed using the Medcalc statistical software.

Categorical variables were compared by the $\chi^2$ test or Fisher's exact test as applicable. P values < 0.05 were considered statistically significant.

[a] LLOQ, lower limit of qualification is 12 IU/mL.

with pan-genotypic antiviral activity for use in patients with renal function impairment. Our study revealed that GLE/PIB could with the SVR rates reaching values of 96.6% and 98.7%, as observed using ITT and PP analysis, respectively in the real-world setting, and there was no significant difference between patients with and without ESRD. The response rates in the

**Table 3. Safety profile of glecaprevir–pibrentasvir therapy.**

| Adverse event, n (%) | All patients (N = 235) | | Non-ESRD (N = 191) | | ESRD (N = 44) | | P-values |
|---|---|---|---|---|---|---|---|
| Pruritus | 77 | (32.9) | 50 | (19.1) | 27 | (62.8) | 0.160 |
| Insomnia | 17 | (7.2) | 10 | (5.2) | 7 | (15.9) | 0.022 |
| Abdominal discomfort | 10 | (4.3) | 9 | (4.7) | 1 | (2.3) | 0.693 |
| GERD | 2 | (0.9) | 2 | (1.0) | 0 | | 1.000 |
| Malaise | 15 | (6.4) | 14 | (7.3) | 1 | (2.3) | 0.315 |
| Dizziness | 8 | (3.4) | 7 | (3.7) | 1 | (2.3) | 1.000 |
| Anorexia | 3 | (1.3) | 2 | (1.0) | 1 | (2.3) | 0.465 |
| Anemia* | | | | | | | |
| G0 | 174 | (74.0) | 164 | (85.9) | 10 | (22.7) | <0.001 |
| G1 | 35 | (14.9) | 15 | (7.9) | 20 | (45.5) | <0.001 |
| G2 | 22 | (9.4) | 9 | (4.7) | 13 | (29.5) | <0.001 |
| G3 | 4 | (1.7) | 3 | (1.6) | 1 | (2.3) | 0.566 |
| Bilirubin | | | | | | | |
| 1.5-3x elevation | 30 | (12.8) | 26 | (13.6) | 4 | (9.1) | 0.418 |
| ≥3x elevation | 3 | (1.3) | 2 | (1.0) | 1 | (2.3) | 0.465 |
| GOT | | | | | | | |
| 3-5x elevation | 2 | (0.9) | 2 | (1.0) | 0 | (0) | 1.000 |
| ≥5x elevation | 2 | (0.9) | 0 | | 1 | (2.3) | 0.187 |
| GPT | | | | | | | |
| 3-5x elevation | 2 | (0.9) | 1 | (0.5) | 1 | (2.3) | 0.340 |
| ≥5x elevation | 1 | (0.4) | 2 | (1.0) | 0 | (0) | 1.000 |

Data are expressed as n/N (%). Categorical variables were compared using the χ2 test or Fisher's exact test. P values <0.05 were considered statistically significant.

*Grade of the side effect according to Common Terminology Criteria for Adverse Events (CTCAE) v4.0.

Table 4. Time course and grade of pruritus.

| Characteristic | All patients (N = 234) | | Non-ESRD (N = 191) | | ESRD (N = 43[a]) | | P value |
|---|---|---|---|---|---|---|---|
| **Grade of pruritus, n (%)*** | | | | | | | |
| G0 | 158 | (67) | 141 | (74) | 17 | (39) | <0.001 |
| G1 | 42 | (18) | 31 | (16) | 11 | (25) | 0.250 |
| G2 | 18 | (8) | 12 | (6) | 6 | (14) | 0.116 |
| G3 | 17 | (7) | 7 | (4) | 10 | (23) | <0.001 |
| **Treatment course of eight weeks** | N = 185 | | N = 155 | | N = 30 | | |
| Skin itch in fourth week, n (%) | 47/185 | (25) | 31/155 | (20) | 16/30 | (53) | <0.001 |
| Skin itch in eighth week, n (%) | 11/185 | (6) | 6/155 | (4) | 5/30 | (17) | 0.018 |
| **Treatment course of twelve weeks** | N = 49 | | N = 36 | | N = 13 | | |
| Skin itch in fourth week, n (%) | 18/49 | (37) | 11/36 | (31) | 7/13 | (54) | 0.184 |
| Skin itch in eighth week, n (%) | 10/49 | (20) | 5/36 | (14) | 5/13 | (39) | 0.104 |
| Skin itch in twelfth week, n (%) | 2/49 | (4) | 1/36 | (3) | 1/13 | (8) | 0.464 |

Data are expressed as n/N (%). Categorical variables were compared using the $\chi^2$ test or Fisher's exact test. P values <0.05 were considered statistically significant.

[a]. One patient left the study due to side effect and therefore, ESRD data are not available for this patient.

*Grade of the side effect according to Common Terminology Criteria for Adverse Events (CTCAE) v4.0.

present cohort were comparable with other clinical trials [2, 19, 20] and recent real-world data [7, 21–24] for patients with and without ESRD. HCV treatment experience from the real world is important because the patient population tends to be clinically more diverse and potentially less adherent to treatment compared with those included in clinical trials. A recent meta-analysis of the real-world effectiveness of GLE/PIB treatment for HCV in patients with chronic HCV infection included 12,531 adults from 18 cohorts with an overall SVR rate of 96.7% in the ITT group [21]. Available data on patients with renal failure is limited, and only 59 patients were included in that study. Our real-world experience is important because we included a substantial number of patients (18%) on dialysis and provides evidence that GLE/PIB can be regarded as a prioritized option for treating patients with hepatitis C regardless of renal status based on its excellent effectiveness and safety.

In terms of safety, majority of the patients (98.7%) in our study were able to complete the scheduled treatment. One patient with advanced breast cancer died of chemotherapy-related infection after achieving ETVR. Two deaths occurred in the ESRD (n = 1) and non-ESRD (n = 1) groups, and both were super-elderly, 80 and 85 years old, respectively. Both the patients died during sleep; the cause of death was suspected to be cardiogenic and related to the HCV treatment. Only one patient in the ESRD group discontinued the scheduled treatment prematurely due to the intolerable side effect of pruritus [1, 2]. We did not observe any evidence of hepatitis B flare-up in our HBV/HCV co-infected patient during the treatment. There were no cases of liver dysfunction leading to interruption of treatment or dose adjustment. The only significant side effect was pruritus, which occurred particularly among patients with ESRD in the first month of treatment. Previous clinical trials [2, 19] and real-world observations [1, 14, 22] have disclosed pruritus as the most frequent adverse event among patients with dialysis, and this side effect appears less frequent in other DAA regimens. Understanding the time course of this side effect is helpful to provide information to help patients overcome the side effect during treatment. After encountering the first patient with premature GLE/PIB therapy discontinuation, we provide proactive measures to educate our patients before initiating therapy and using medications such as antihistamine and/or topical agent to relieve pruritus

symptom therapy initiation. Such measures have helped patients adhere +to the GLE/PIB therapy in previous studies [25, 26].

The present study has several limitations. First, there may be reporting bias with regard to the side effects because our study was retrospective, and therefore, mild side effects and the prevalence of pruritus before treatment between groups may not have been recorded in clinical practice. Because one third of our patients reported pruritus as a side effect in our initial treatment experience, we only recorded and graded this side effect; therefore, other side effects such as fatigue or change in appetite were not recorded. Second, 84% of our patients were treatment-naïve with either genotype 1 or 2, and so our study population was very homogenous. More experience is required to confirm the treatment result of GLE/PIB in patients with e.g. less common genotypes. Third, because we included patients from a single tertiary center, the treatment outcome at community hospitals or the clinics should be further confirmed. Fourth, we did not assess the on-treatment HCV RNA test because the Taiwan NHI only reimburses HCV RNA testing for SVR and ETVR. We were not able to provide viral kinetic data regarding the six patients not achieving ETVR and three patients without SVR for further analysis. Fifth, our study includes only naïve patients undergoing DAA treatment. Therefore, we cannot extrapolate the efficacy data of GLE/PIB to patients who have undergone prior failed DAA therapy.

## Conclusion

In this real-world study, we demonstrated that GLE/PIB-based therapy is highly effective for patients with and without ESRD.

## Supporting information

**S1 Data.**
(PDF)

## Author Contributions

**Conceptualization:** Hsu-Heng Yen, Pei-Yuan Su.

**Data curation:** Hsu-Heng Yen, Pei-Yuan Su, Ya-Huei Zeng, I-Ling Liu, Siou-Ping Huang, Yu-Chun Hsu, Chia-Wei Yang.

**Formal analysis:** Hsu-Heng Yen, I-Ling Liu, Siou-Ping Huang, Yu-Chun Hsu.

**Funding acquisition:** Hsu-Heng Yen.

**Investigation:** Hsu-Heng Yen.

**Project administration:** Yu-Chun Hsu.

**Resources:** Pei-Yuan Su, Yu-Chun Hsu.

**Supervision:** Yang-Yuan Chen, Shun-Sheng Wu, Kun-Ching Chou.

**Writing – original draft:** Hsu-Heng Yen.

**Writing – review & editing:** Hsu-Heng Yen, Pei-Yuan Su, Ya-Huei Zeng, I-Ling Liu, Siou-Ping Huang, Yu-Chun Hsu, Yang-Yuan Chen, Chia-Wei Yang, Shun-Sheng Wu, Kun-Ching Chou.

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
