## [Decision Letter · Decision Letter 0]

1 Jul 2020

PONE-D-20-18097

Glecaprevir–Pibrentasvir for Chronic Hepatitis C: Comparing treatment effect in patients with and without end-stage renal disease in a real-world setting

PLOS ONE

Dear Dr. Hsu-Heng Yen,

Thank you for submitting your manuscript to PLOS ONE. After careful consideration, we feel that it has merit but does not fully meet PLOS ONE’s publication criteria as it currently stands. Therefore, we invite you to submit a revised version of the manuscript that addresses the points raised during the review process.

We look forward to receiving your revised manuscript.

Kind regards,

Tatsuo Kanda, M.D., Ph.D.

Academic Editor

PLOS ONE

Journal Requirements:

2. Our internal editors have looked over your manuscript and determined that it is within the scope of our Liver Diseases Call for Papers. This collection of papers is headed by a team of Guest Editors for PLOS ONE. Additional information can be found on our announcement page: https://collections.plos.org/s/liver-diseases. If you would like your manuscript to be considered for this collection, please let us know in your cover letter and we will ensure that your paper is treated as if you were responding to this call. If you would prefer to remove your manuscript from collection consideration, please specify this in the cover letter.

3.During our internal evaluation of the manuscript, we found minor instances of text overlap between your submission and the following previously published works:

https://onlinelibrary.wiley.com/doi/abs/10.1111/jvh.13265

Please revise the manuscript to rephrase the duplicated text and cite your sources.

4. Thank you for stating in the text of your manuscript " documentation of informed consent was waived because the study was conducted retrospectively". Please also add this information to your ethics statement in the online submission form.

Reviewers' comments:

Reviewer's Responses to Questions

**Comments to the Author**

1. Is the manuscript technically sound, and do the data support the conclusions?

Reviewer #1: Yes

Reviewer #2: Partly

2. Has the statistical analysis been performed appropriately and rigorously? 

Reviewer #1: Yes

Reviewer #2: Yes

3. Have the authors made all data underlying the findings in their manuscript fully available?

Reviewer #1: Yes

Reviewer #2: Yes

4. Is the manuscript presented in an intelligible fashion and written in standard English?

Reviewer #1: Yes

Reviewer #2: Yes

5. Review Comments to the Author

Reviewer #1: 1. Please mention the reason why skin itching was higher in ESRD patients. Were they actually treatment-related or present before treatment?

2. Were all patients treated at the same department in the hospital or were ESRD patients treated at a different department?

3. Please mention the reason for death in 3 patients.

4. Please describe the method of HCV genotyping and HCV quantification.

5. Was there any possible reason why HCV RNA level in ESRD patients was lower than others?

6. Please include the treatment history in Table 1 and mention the treatment efficacy according to treatment history and treatment duration.

7. Please mention the presence of treatment history in relapsed patients. Were the drug resistance associated variants examined in these patients?

8. Line 52 “with oral DAAs”, there is error in font style.

Reviewer #2: In this article, the author retrospectively evaluated treatment efficacy and safety of GLE/PIB in patients with HCV and end-stage renal disease. They showed similar SVR rate and higher prevalence of pruritus among patients with ESRD.

1) Treatment regimen should be described in more detail. The way how to decide treatment duration in Taiwan should be mentioned.

2) The number of patients who have experienced treatment with DAAs should be described.

3) The relationship between death and treatment should be assessed.

4) The complications other than pruritus should be described.

6. PLOS authors have the option to publish the peer review history of their article (what does this mean?). If published, this will include your full peer review and any attached files.

Reviewer #1: No

Reviewer #2: No

---

## [Author Response · Author response to Decision Letter 0]

6 Jul 2020

Reviewer: 1

1. Please mention the reason why skin itching was higher in ESRD patients. Were they actually treatment-related or present before treatment?

Response: Thank for your comment. As mentioned in the discussion, previous clinical trials [2, 18] and real-world observations [1, 13, 21] have disclosed pruritus as the most frequent adverse event among patients with dialysis. Itching is constantly observed as the most common side effect for GLI/PIB based therapy. Our study is the first to describe the time course change of pruritus symptom. We found the majority patients experienced pruritus during the first four week of treatment (Table 4) and we believe it should be treatment related. We added Table 3 to describe other side effects of GLI/PIB therapy in our present study.

2. Were all patients treated at the same department in the hospital or were ESRD patients treated at a different department?

Response: Thank for your comment. In Taiwan, the use of DAA was restricted to licensed gastroenterologist and all the patients included treatment in the same department during the treatment period. 

3. Please mention the reason for death in 3 patients.

Response: Thank for your comment. Unlike clinical trial, our report of real-world experience reflects the true patients. The cause of death was discussed in the discussion section in the revised manuscript. Two deaths occurred in the ESRD (n = 1) and non-ESRD (n = 1) groups, and both were super-elderly, 80 and 85 years old, respectively. Both patients died during sleep and had no signs of other systemic disease during the last phone contact. Cardiogenic cause of death was suspected and judged not to be related to the HCV treatment. The third patient died of chemotherapy related infection. 

4. Please describe the method of HCV genotyping and HCV quantification.

Response: Thank for your comment. We used ART HCV assays (RealTime HCV and HCV Genotype II, Abbott Molecular, Abbott Park, IL, USA) to quantitatively measure HCV RNA concentrations and genotyping. We addressed this point in the revised manuscript. 

5. Was there any possible reason why HCV RNA level in ESRD patients was lower than others?

Response: Thank for your comment. Previous reports (Kaiser T, et al. Kinetics of hepatitis C viral RNA and HCV-antigen during dialysis sessions: evidence for differential viral load reduction on dialysis. J Med Virol. 2008;80(7):1195-1201 ; Fabrizi F, et al. Kinetics of hepatitis C virus load during hemodialysis: novel perspectives. J Nephrol. 2003;16(4):467-475) have described a variable reduction of HCV-RNA during hemodialysis treatment sessions. Various mechanisms have been postulated such as adsorption of HCV onto dialysis membrane, HCV escape into spent dialysate, or destruction of HCV particles during dialysis.

6. Please include the treatment history in Table 1 and mention the treatment efficacy according to treatment history and treatment duration.

Response: Thank for your comment. We agree with your opinion that prior DAA treatment history is important to guide subsequent DAA therapy. As we mentioned efficacy and safety evaluation section, GLI/PIB was reimbursed by the National Health Insurance of Taiwan since Aug 2018 for patients with confirmed HCV viremia without prior use of DAA. Thus, there is no relevant treatment history for patients included in this analysis. We include the interferon therapy history in the revised Table 1. We will acknowledge it as our study limitation.

7. Please mention the presence of treatment history in relapsed patients. Was the drug resistance associated variants examined in these patients?

Response: Thank for your comment. We included only DAA treatment naïve patients in this study. Drug resistance is not tested prior to Glecaprevir–Pibrentasvir therapy in this study.

8. Line 52 “with oral DAAs”, there is error in font style.

Response: Thank for your comment. We correct this error in revised manuscript. 

Reviewer #2: In this article, the author retrospectively evaluated treatment efficacy and safety of GLE/PIB in patients with HCV and end-stage renal disease. They showed similar SVR rate and higher prevalence of pruritus among patients with ESRD.

1) Treatment regimen should be described in more detail. The way how to decide treatment duration in Taiwan should be mentioned.

Response: Thank for your comment. GLI/PIB was reimbursed by the National Health Insurance of Taiwan since Aug 2018 for patients with confirmed HCV viremia without prior use of DAA. The treatment period was 8 weeks for patients without liver cirrhosis and 12 weeks for patients with compensated liver cirrhosis. The treatment period was 16 weeks for genotype 3 patients with prior interferon therapy 

2) The number of patients who have experienced treatment with DAAs should be described.

Response: Thank for your comment. In Taiwan, the use of DAA was restricted to licensed gastroenterologist and all the patients included treatment in the same department during the treatment period. Patients with resistance to DAA therapy would be referred to special hepatology center and thus, we include only treatment naïve patients in this study. We will acknowledge it as our study limitation.

3) The relationship between death and treatment should be assessed.

Response: The cause of death was discussed in the discussion section in the revised manuscript. Two deaths occurred in the ESRD (n = 1) and non-ESRD (n = 1) groups, and both were super-elderly, 80 and 85 years old, respectively. Both patients died during sleep and had no signs of other systemic disease during the last phone contact. Cardiogenic cause of death was suspected and judged not to be related to the HCV treatment. The third patient died of chemotherapy related infection.

4) The complications other than pruritus should be described.

Response: Thank for your comment. We add Table 3 to mention other complications associated with GLI/PIB therapy in the revised manuscript.

---

## [Decision Letter · Decision Letter 1]

23 Jul 2020

PONE-D-20-18097R1

Glecaprevir–Pibrentasvir for Chronic Hepatitis C: Comparing treatment effect in patients with and without end-stage renal disease in a real-world setting

PLOS ONE

Dear Dr. Hsu-Heng Yen,

Thank you for submitting your manuscript to PLOS ONE. After careful consideration, we feel that it has merit but does not fully meet PLOS ONE’s publication criteria as it currently stands. Therefore, we invite you to submit a revised version of the manuscript that addresses the points raised during the review process.

We look forward to receiving your revised manuscript.

Kind regards,

Tatsuo Kanda, M.D., Ph.D.

Academic Editor

PLOS ONE

Reviewers' comments:

Reviewer's Responses to Questions

**Comments to the Author**

1. If the authors have adequately addressed your comments raised in a previous round of review and you feel that this manuscript is now acceptable for publication, you may indicate that here to bypass the “Comments to the Author” section, enter your conflict of interest statement in the “Confidential to Editor” section, and submit your "Accept" recommendation.

Reviewer #1: (No Response)

Reviewer #2: All comments have been addressed

2. Is the manuscript technically sound, and do the data support the conclusions?

Reviewer #1: (No Response)

Reviewer #2: (No Response)

3. Has the statistical analysis been performed appropriately and rigorously? 

Reviewer #1: (No Response)

Reviewer #2: (No Response)

4. Have the authors made all data underlying the findings in their manuscript fully available?

Reviewer #1: (No Response)

Reviewer #2: (No Response)

5. Is the manuscript presented in an intelligible fashion and written in standard English?

Reviewer #1: (No Response)

Reviewer #2: (No Response)

6. Review Comments to the Author

Reviewer #1: Because pruritus is common in dialysis patients, showing the prevalence of pruritus before treatment between groups will help to clearly understand the results of time course data in table 4.

Reviewer #2: (No Response)

7. PLOS authors have the option to publish the peer review history of their article (what does this mean?). If published, this will include your full peer review and any attached files.

Reviewer #1: No

Reviewer #2: No

---

## [Author Response · Author response to Decision Letter 1]

23 Jul 2020

Reviewer #1: Because pruritus is common in dialysis patients, showing the prevalence of pruritus before treatment between groups will help to clearly understand the results of time course data in table 4.

Response: Thank for your comment. We believe this issue is important as the reviewer mentioned. The presence and the severity of pruritus at baseline is not a contra-indication for anti-HCV therapy in our real-world practice. As this is a retrospective real-world study, we didn’t record the pruritus symptoms before treatment initially in our practice. As mentioned in the discussion, after encountering the first patient with premature GLE/PIB therapy discontinuation from this side effect, we learned the importance of providing proactive measures to educate our patients before initiating therapy and using medications such as antihistamine and/or topical agent to relieve pruritus symptom therapy initiation. Such measures have helped patients adhere to the GLE/PIB therapy in previous studies. We acknowledge this as a limitation of the present study in the revised manuscript.

---

## [Editor Report · Decision Letter 2]

27 Jul 2020

PONE-D-20-18097R2

Glecaprevir–Pibrentasvir for Chronic Hepatitis C: Comparing treatment effect in patients with and without end-stage renal disease in a real-world setting

PLOS ONE

Dear Dr. Hsu-Heng Yen,

Thank you for submitting your manuscript to PLOS ONE. After careful consideration, we feel that it has merit but does not fully meet PLOS ONE’s publication criteria as it currently stands. Therefore, we invite you to submit a revised version of the manuscript that addresses the points raised during the review process.

ACADEMIC EDITOR: 

1) Authors should describe the definition of ESRD clearly.

We look forward to receiving your revised manuscript.

Kind regards,

Tatsuo Kanda, M.D., Ph.D.

Academic Editor

PLOS ONE

---

## [Author Response · Author response to Decision Letter 2]

27 Jul 2020

Response to ACADEMIC EDITOR’s comments 

1) Authors should describe the definition of ESRD clearly. 

Response: Thank for your comment to improve the manuscript. In the present study, we define ESRD as CKD Stage 5D who received dialysis (Kanda T, et al. Hepatol Int. 2019;13(2):103-9. )

---

## [Editor Report · Decision Letter 3]

30 Jul 2020

Glecaprevir–Pibrentasvir for Chronic Hepatitis C: Comparing treatment effect in patients with and without end-stage renal disease in a real-world setting

PONE-D-20-18097R3

Dear Dr. Hsu-Heng Yen,

We’re pleased to inform you that your manuscript has been judged scientifically suitable for publication and will be formally accepted for publication once it meets all outstanding technical requirements.

Kind regards,

Tatsuo Kanda, M.D., Ph.D.

Academic Editor

PLOS ONE
---

## [Editor Report · Acceptance letter]

4 Aug 2020

PONE-D-20-18097R3 

Glecaprevir–Pibrentasvir for Chronic Hepatitis C: Comparing treatment effect in patients with and without end-stage renal disease in a real-world setting 

Dear Dr. Yen:

I'm pleased to inform you that your manuscript has been deemed suitable for publication in PLOS ONE. Congratulations! Your manuscript is now with our production department. 

Kind regards, 

on behalf of

Dr. Tatsuo Kanda 

Academic Editor

PLOS ONE